# Positive Youth Identity: The Role of Adult Social Support

Agnieszka Jankowska-Tvedten [1] and Nora Wiium [2,*]

[1]  Department of Mental Health and Substance Abuse, Randaberg Municipality, 4070 Randaberg, Norway; a.jankowska@hotmail.co.uk

[2]  Department of Psychosocial Science, Faculty of Psychology, University of Bergen, 5020 Bergen, Norway

*   Correspondence: nora.wiium@uib.no

**Abstract:** Positive identity is essential to adolescents' well-being and a successful transition to adulthood. Understanding factors that contribute to identity formation is important to promote healthy development. The present study applies the Positive Youth Development (PYD) framework to explore associations between adult social support and positive identity among Norwegian high school students (N = 220, $M_{age}$ = 17.30, SD = 1.12). Reflecting on the ecology of adolescence, the study examined the extent to which family, other significant adults, and school support relate to different dimensions of positive identity (personal power, self-esteem, sense of purpose, and positive view of future). Further, the hierarchy of importance among the support variables was investigated. Cross-sectional data on items measuring the different dimensions of social support and positive identity were analyzed. Findings from regression analyses indicated that adult support in all three ecological contexts (family, neighborhood, and school) was significantly and positively related to positive identity and at least one of its four dimensions ($\beta$ = 0.15–0.27, $p < 0.05$). Support from other significant adults emerged as the most important predictor, followed by family and then school support. While future research should also consider support accessed through social media, our findings extend PYD scholarship in the Norwegian context by highlighting the importance of key youth ecological contexts when fostering positive youth identity.

**Keywords:** positive youth development; developmental assets; identity development; adult social support; Norway

## 1. Introduction

Positive identity, such as having control over one's future, holding a positive view of oneself, and being satisfied with one's life, is an important marker of positive development and thriving in adolescence [1,2]. Identity has been described as a cognitive structure that incorporates one's answers to the questions about who they are and what they want out of life [3]. Although awareness of self has its foundations in early childhood, adolescence is identified as the developmental period in which a coherent sense of self is formed [4]. Without developing a cohesive sense of identity in adolescence, milestones like establishing intimate relationships in early adulthood, instigating generative projects in midlife, and growing old with a sense of integrity and accomplishment will likely be out of reach [5].

Adolescence is a transitional period from childhood to adulthood. Embedded within this period of life are multiple changes at biological, social, and cognitive levels that prepare the individual for adulthood as adolescents increase their autonomy and become more independent from their parents [6]. Moreover, they begin to stand up to external expectations and new societal demands to prepare for their future lives as adults. Examples include completing school or making occupational choices [4]. Very few developmental periods are characterized by so many changes at so many different levels [7]. Thus, adolescents are particularly vulnerable, and their healthy development depends on guidance and support from good role models in their immediate context [8].

Research has consistently shown that a positive identity is essential to adolescents' well-being and a successful transition to adulthood [3,9]. In factors fostering positive identity development, the importance of supportive adult relationships within key youth ecological contexts has been highlighted [10,11]. Indeed, in line with Benson [12], after long years of extensive studies, scholars and practitioners have come up with a surprisingly simple, but powerful conclusion: "nothing—nothing—has more impact in the life of a child than positive relationships" (p. 13).

Understanding and, ultimately, fostering positive identity development during adolescence is crucial as it can guide researchers and practitioners in promoting positive developmental trajectories as young people transition into adulthood. The current study adopts the Positive Youth Development (PYD) framework to investigate the role of adult social support on positive identity among adolescents in the Norwegian context. Further, it explores the relative importance of adult support within the contexts of family, other significant adults (neighborhood), and school.

## 1.1. The Norwegian Sociocultural Context

In May 2023, there was approximately 323,179 youth aged 16–20 years in Norway. That number represents 5.9% of the 5.5 million inhabitants [13]. Norway has a well-developed welfare and social security system. Education from primary school up to university level is free. These core values are based on UNESCO's Salamanca Declaration [14]. Most adolescents attend public high schools, giving schools a unique position to facilitate positive development [15]. The Dream School Program and "Public Health and Life Skills" subject in Norway are examples of school policies aimed at positive adolescent development. The goal of the Dream School Program is to create a learning environment where students can build their confidence, stimulate positive development, and experience a sense of belonging [16]. Similarly, the "Public Health and Life Skills" subject taught in schools is designed to help students improve resilience and self-regulation and to teach them how to cope with failure, rejection, and adversity. Both programs intend to support students in their life choices and help them develop positive identities and self-esteem [15].

Various national statistical data portray a picture of how satisfied and content Norwegian adolescents are. The surveys show that most adolescents in Norway thrive and report feeling happy [17]. Quality of life findings also indicate that almost 90% of teenagers are satisfied with their life. Boys are generally more satisfied with life than girls of the same age [17]. Statistically, 6 out of 10 boys display higher optimism about the future than only 42% of girls. Similarly, 7 out of 10 boys can cope with upcoming issues, whereas every fourth girl shares the same feeling [17]. Data collection in 2020 and 2021 occurred after the coronavirus pandemic in Norway.

Nevertheless, when studied closely, the research shows that scores on Norwegian youth's self-esteem are not as high, especially in early and middle adolescence [18]. For example, on a scale from 1 (low) to 4 (high), the level of self-esteem for females was 2.8 and was 3.1 for males at the age of 15 [19]. This could potentially be explained by the parallel increase in psychological distress in that period, which is more frequent among girls than boys [20].

Less is known about the positive and influencing role of adult social support on thriving and positive development among adolescents in Norway. Considering that identity development is a complex process that involves both the person and society, the Norwegian context is relevant for study. Although there are solid grounds for satisfaction in what has already been achieved through the support and opportunity provided to young people in Norway, areas of policy and practice can be improved [21], with insights from the PYD perspective.

## 1.2. Positive Youth Development, Developmental Assets, Social Support, and Positive Identity

The turn of the 21st century saw a surge of research focused on the positive view of human development. In adolescent literature, PYD, as opposed to the deficit-based approach

(that highlights youth weaknesses and problems), addresses youth strengths, assets, and environmental opportunities that can support healthy development and thriving [22]. This strengths-based approach focuses on promoting factors that make people healthier instead of inhibiting factors that cause people to fall sick [10]. Importantly, for the present study, PYD perceives healthy development in terms of achieving developmental tasks, such as a coherent sense of identity [23]. Rooted in the relational developmental systems theory, PYD highlights the plasticity of human development, where a dynamic interaction between youth and their contexts (e.g., home and school) can lead to constructive and systematic changes in their development [24,25].

The present research is based on the Developmental Assets Framework (DAF), one of the most influential PYD models [26]. Coining the term developmental assets, Benson and his colleagues [27] describe the relationships, opportunities, and character strengths firmly associated with adolescents' well-being. Thus, developmental assets are the building blocks young people need to grow into healthy, caring, and responsible adults [28].

Consistent with the DAF model, there are 40 developmental assets in total, comprising 20 internal and 20 external assets [1]. The external assets represent positive experiences and qualities that parents, school, friends, and the local community contribute to young people and consist of four subcategories: support (indicating caring, appreciation, and acceptance from people in youth contexts); empowerment (signifying how the context provides youth with a feeling of being valuable, safe, and respected); boundaries and expectations (designating clear rules and consequences for behavior, good role models, encouragement, and expectations of responsible behavior); and constructive use of time (representing the opportunities afforded to interact with peers and adults in leisure activities and to learn new skills) [1,8,29].

Similarly, the internal assets reflect individual and psychological qualities of youth and are also divided into four subcategories: commitment to learning (reflecting understanding the importance of learning and believing in one's abilities); positive values (indicating developing values that can facilitate good life choices), social competencies (which is the ability to effectively interact with others, make choices, and master new situations); and positive identity (signifying believing in self-value and feeling of control over life) [1,8,29]. Experiencing more assets in the asset category (vertical pile-up) and assets across different asset categories (horizontal stacking) is associated with adolescents' increased well-being and thriving [29]. In line with Benson [1], thriving occurs when young people show signs of leadership, can resist danger or overcome adversity, are willing to help others, value diversity, delay gratification, maintain good health, and have school success.

Adopting the DAF model and PYD framework, the present study aimed to assess associations in a sample of Norwegian adolescents by examining the link between an external (social support) and internal (positive identity) developmental asset category. Although scholars propose the integrative approach, where all the asset categories are considered in the study of youth development, each category can be treated as independent due to its unique effect on youth developmental outcomes [25]. In our study, we are interested in how different dimensions of social support influence adolescent positive identity. Thus, we focus on these two asset categories and not the cumulative assumption of the DAF model. Together with the other developmental assets, these two asset categories can facilitate thriving and, eventually, contribution among young people [2,30].

*1.3. Empirical Evidence on the Association between Social Support and Adolescent Identity*

Extensive cross-sectional and longitudinal studies conducted primarily in the United States, together with a growing body of international research, continue to validate the utility of developmental assets in facilitating positive developmental outcomes in children, adolescents, and emerging adults [2,10,28,31]. Despite such empirical interest, the research investigating relations between support and positive identity as defined by the DAF model is limited. From the limited literature, Kosic et al. [32] confirmed the significant role of social support in boosting internal assets (positive identity and social competence), and

positive outcomes, such as academic achievement among ethnic Italian youth (N = 524). The study provided valuable evidence that supportive relationships within schools, families, and communities can foster positive youth development. In another study comparing developmental assets in four European countries, Dost-Gözkan and colleagues [2] examined the associations between external assets (support, empowerment, boundaries, and expectations) and positive identity (an internal asset). A total of 2055 young adults from Norway, Romania, Slovenia, and Turkey participated in the study. Similar results were found in the four countries, with external assets being positively linked to positive identity.

Due to limited research on the current topic within PYD, we also considered relevant studies that have examined the role of social support in youth development outside the PYD framework. Given that young people immerse themselves in widening circles of social interactions during adolescence, this literature review focuses on three core developmental settings: the family, the school, and the neighborhood/local community [33]. Extensive research indicates that supportive parents and other adults in the family are critical for positive development in adolescence [34,35]. For example, Trost et al. [35] provided evidence suggesting that some aspects of the family context (e.g., democratic parenting, parental warmth, and communication) and adolescents' interactions with parents were essential for identity development and adjustment in 13–14-year-old Swedish adolescents (N = 3667). A related study with 239 Finnish students that examined adolescent subjective well-being and family dynamics supported the important role of parental involvement and communication [34].

In addition to the family context, literature on adult support in the school milieu has also demonstrated their significance in promoting positive youth development [36,37]. Tian et al. [36] found that students who felt supported by teachers demonstrated higher achievement and subjective well-being at school. This longitudinal study consisted of 1316 high school students aged 11–19 living in southern China.

With increasing age, the neighborhood and local community evolve as a different ecological niche in which adolescents spend time. Like the school setting, the community context is particularly significant because it can influence young people [38]. A study by Theokas and colleagues [38] revealed that young people who experienced caring and support from non-parental adults in their neighborhood scored higher on indicators of thriving, such as a positive view of the self and high expectations for achievement. The study used data from 222,596 youth from public schools across the United States. Participants were between the ages of 10 and 19.

The empirical research within and outside the PYD perspective consistently provides strong evidence for an association between adult social support and a positive sense of self among adolescents in diverse cultural contexts. The findings also support the notion that adolescents' development needs to be explored from a systematic point of view, including the multiple ecological contexts in which youth live, learn, and grow [27], for example, in the family, school, neighborhood, and the local community. However, whether and how developmental relationships and their support influence positive identity among adolescents is not entirely clear in a Norwegian context, a typical individualistic cultural setting that encourages autonomy among its population, including the youth. It is possible that the expectation of autonomy can influence Norwegian youth's perception and seeking of support. Since there is cultural diversity in perceived support [39], further studies are needed in Norway.

*1.4. Aims of the Present Study*

International research on social support and positive identity provides theoretical and empirical grounds for investigating positive identity and dimensions of adult social support. This paper aims to study if similar associations can be found in a sample of Norwegian high school students. More specifically, we studied how support can influence positive identity in an individual–context interaction within the DAF model and PYD Framework. In line with the DAF model, the current study operationalized social support as support and

affection from the family, parental involvement in schooling, caring neighborhoods, and schools with a culture of encouragement [28]. Thus, the concept of support is investigated separately for family, other significant adults, and school to better understand the influence of each context. Positive identity is framed as a PYD self-concept and indicates perceiving control over one's life and future, having a purpose in life, feeling good about one's future, and having high self-esteem [1]. Moreover, demographic factors, such as age, gender, and parents' educational background, have some implications for the report of the assets [31] and thus are controlled in the current study.

The following hypotheses were formed: (1) Based on earlier findings on the link between social support and positive identity, Hypothesis 1 states that social support will be positively and significantly associated with positive identity among Norwegian adolescents. (2) Empirical findings show that family, other significant adults, and school support have differential impact on positive youth outcomes. As such, Hypothesis 2 states that dimensions of social support will be positively and significantly associated with positive identity and that family, other significant adults, and school support factors will vary in predicting positive identity among Norwegian adolescents.

The current study is of great importance, considering that adolescent years are critical for building successful foundations of health and well-being for subsequent stages of life [1,3,24]. Identifying factors that can promote positive identity that will enable youth to successfully take on adult roles and, consequently, contribute meaningfully to the self and to society is crucial for research and application. Moreover, understanding the differential contributions of family, school, and other adults' support towards positive identity can help in the formulation of policies that can embrace the complimentary role of support in all three contexts in practice.

## 2. Materials and Methods

### 2.1. Participants

The current study used data collected for a cross-national project on PYD [40]. Our sample consisted of 220 students attending public high schools in Eastern and Western Norway. Participants were selected through convenience sampling and their ages varied from 16 to 20 years ($M_{age}$ = 7.30, SD = 1.12; 47.7% girls). More than half of the participants reported the highest education of their parents as college or university, with 55% having a father with higher education and 68.6% having a mother with a similar educational level.

### 2.2. Measures

Social support and positive identity were measured using the Developmental Assets Profile questionnaire [41]. Social support, as an external asset, was used as the independent variable. A seven-item scale reflecting three dimensions of support (family, other significant adults, and school support) was used to assess how youth perceived adults as supportive. Family support was measured using four items covering parental support, family communication, and family involvement. Sample items are "I ask my parents for advice" and "I have parent(s)/guardian(s) who try to help me succeed". Other significant adults' support was measured using two items covering non-parental adults and neighbor support. A sample item is "I have support from adults other than my parents". School support was measured using one item: "I have a school that cares about students and encourages them". Responses were coded on a 4-point Likert scale from 1 (not at all or rarely always) to 4 (extremely or almost). A high score indicated high levels of social support. Cronbach's alphas of the variables, ranging from 0.51 to 0.85, were in the range of a poor level to a very good level and were comparable to reports given in previous studies [10,42]. The poor level of reliability of other significant adults' support is most likely due to the use of two items in its assessment.

Positive identity, as an internal asset, was used as the dependent variable. A six-item scale reflecting four dimensions of positive identity assessed participants' personal power, self-esteem, a sense of purpose, and a positive view of the future. Personal power was

assessed through three items referring to the feeling of control over things. A sample item is: "I deal with disappointment without getting too upset". Self-esteem was assessed using one item: "I feel good about myself". Sense of purpose was assessed using one item: "I am thinking about my purpose in life". A positive view of the future was evaluated by one item covering feeling optimistic about the future. Responses were coded on a 4-point Likert scale from 1 (not at all or rarely always) to 4 (extremely or almost). Cronbach's alpha of the positive identity asset category was 0.83, while Cronbach's alpha of the personal power dimension was 0.76. The Cronbach's alphas of the current study are acceptable and comparable to those reported in previous studies [10,42].

For demographic variables, participants were asked to provide information about their age, gender (male or female), and parents' educational level (i.e., no education, primary school, high school, technical or vocational school, and university).

### 2.3. Procedure

Data collection took place in 2019. An e-mail about the research and its purpose was sent to principals of the selected high schools requesting participation in the study. After the permission was granted, informed consent was sought from students who were advised about the study's purpose and procedures. Data collection, which took approximately 30–45 min, was conducted during school hours through SurveyXact, an online platform. To avoid language misunderstandings, the questionnaire, initially developed in English, was translated into Norwegian by Semantix Translations Norway AS, an authorized company that offers translation and interpreting services. The study was approved by the NSD—Norwegian Center for Research Data (51708/3/IJJ).

### 2.4. Data Analysis

Data analyses were performed using the IBM Statistical Package for the Social Sciences (IBM SPSS version 27). G*Power 3 [43] was used to conduct a power analysis to determine the sample size that allowed for the assessment of meaningful associations and the detection of effect sizes (small, medium, or large). Missing cases varied from 0% with gender to 10.5% with an item on positive identity, "I feel good about my future". Pairwise deletion was used to handle missing cases. The procedure excludes participants from the analysis when data are missing and includes them when data are available [44]. Preliminary analyses were carried out to verify the data's linearity, homoscedasticity, and normal distribution. The skewness and kurtosis of all studied variables were within the acceptable range of −2 to +2 and −7 to +7, respectively [45]. For example, skewness and kurtosis for social support were 0.6 and 0.5, respectively. Reliability tests were conducted to check the internal consistencies of scale items.

To assess the distribution and the extent to which participants reported the study variables, frequency and descriptive analyses were performed (Table 1). Participants' age was re-coded to 1 (middle adolescence, 16–17 years old) and 2 (late adolescence, 18–20 years old) to examine how middle adolescents differed from late adolescents in their responses. Similarly, parents' educational level was re-coded as 1 (high school or lower), and 2 (vocational, technical, or university) to assess how participants with parents with high and low education differed in their responses. Gender was coded as 1 (boy) and 2 (girl).

Furthermore, correlation analysis was carried out to examine the strength of associations between support with its three dimensions and positive identity with its four dimensions. Within the Developmental Framework, asset categories are usually analyzed as composite variables. The current study goes further to examine which dimensions are the strongest predictors. Thus, 10 sets of regression analyses were conducted to assess the influence of support with its three dimensions on positive identity along with its four dimensions.

**Table 1.** Study variables among high school students in Norway.

| Study Variable | Item | Range | Mean (SD) |
|---|---|---|---|
| Age | How old are you? (16–17 or 18–20) | 1–2 | 1.43 (0.50) |
| Gender | What is your gender? (Male or Female) | 1–2 | 1.48 (0.50) |
| Father's education | Secondary school or lower versus vocational, technical, polytechnic or university | 1–2 | 1.83 (0.37) |
| Mother's education | Secondary school or lower versus vocational, technical, polytechnic or university | 1–2 | 1.87 (0.33) |
| Social support (7 items; $\alpha$ = 0.83) | | 1–4 | 3.04 (0.59) |
| Family support (4 items; $\alpha$ = 0.85) | Sample item: I have a family that gives me love and support. | 1–4 | 3.34 (0.66) |
| Other significant adults (2 items; $\alpha$ = 0.51) | Sample item: I have good neighbours who care about me. | 1–4 | 2.47 (0.77) |
| School (1 item) | Sample item: I have a department that cares about students and encourages them. | 1–4 | 2.92 (0.75) |
| Positive identity (6 items; $\alpha$ = 0.84) | | 1–4 | 2.89 (0.65) |
| Personal power (3 items; $\alpha$ = 0.76) | Sample item: I feel I have control of my life and future. | 1–4 | 2.82 (0.72) |
| Self-esteem (1 item) | Sample item: I feel good about myself. | 1–4 | 3.02 (0.89) |
| Sense of purpose (1 item) | Sample item: I am thinking about what my purpose is in life. | 1–4 | 2.79 (0.95) |
| Positive view of future (1 item) | Sample item: I feel good about my future. | 1–4 | 3.04 (0.89) |

Note. $\alpha$ = Cronbach's alpha; SD = Standard Deviation.

For five sets of models, at Step 1, the demographics (age, gender, and parents' educational background) were included as control variables, and at Step 2, the social support asset category as a composite mean score was included as an independent variable. For the next five sets, at Step 1, the demographics were included as control variables, and at Step 2 the three dimensions of social support as independent variables were introduced. The dependent variables were positive identity and its four dimensions: personal power, self-esteem, sense of purpose, and positive view of future. To test the hypothesis that ecological contexts of social support (i.e., family, other significant adults, and school) will vary in predicting positive identity among Norwegian adolescents, the Pratt index was estimated. A predictor variable's Pratt index is computed based on the variable's β-weight, its correlation with the outcome variable, and the total $R^2$ in the regression model, d = (β × rxy)/$R^2$ [46].

**3. Results**

*3.1. Descriptive and Correlation Analyses*

Results on the mean scores (range: 1–4), indicate moderate scores for social support 3.04 (SD = 0.59) and positive identity 2.89 (SD = 0.65). For mean scores on the dimensions, see Table 1. Correlation analyses in Table 2 showed statistically significant associations between all main study variables, with medium to strong correlations for most pairs [47], except for the correlation between school support and sense of purpose (r = 0.10, ns). A strong correlation was found between the composite variables, social support, and positive identity (r = 0.52, *p* < 0.01), suggesting a covariation between the occurrence of social support and a higher level of positive identity. Amongst social support's three dimensions and the dimensions of positive identity, significant, but weak to moderate correlations (except for the non-significant correlation reported above) were observed, ranging from 0.15 to 0.46. The strongest correlation was between other significant adults' support and positive view of future (r = 0.46, *p* < 0.01) and the weakest between other significant adults' support and sense of purpose (r = 0.15, *p* < 0.05).

**Table 2.** Correlation analysis among demographic variables, social support, and positive identity.

| Study Variables | 2. | 3. | 4. | 5. | 6. | 7. | 8. | 9. | 10. | 11. | 12. | 13. |
|---|---|---|---|---|---|---|---|---|---|---|---|---|
| 1. Age | 0.02 | −0.12 | −0.07 | −0.12 | −0.06 | −0.16 * | −0.16 * | −0.09 | −0.10 | −0.06 | 0.06 | −0.12 |
| 2. Gender | - | −0.08 | −0.08 | 0.07 | 0.04 | 0.07 | 0.09 | −0.17 * | −0.18 ** | −0.17 * | −0.06 | −0.08 |
| 3. Father's education | | - | 0.22 ** | 0.01 | 0.01 | −0.01 | 0.03 | 0.03 | 0.04 | 0.06 | −0.04 | 0.04 |
| 4. Mother's education | | | - | 0.08 | 0.07 | 0.06 | 0.05 | 0.14 | 0.14 | 0.15 * | 0.01 | 0.11 |
| 5. Social Support | | | | - | 0.92 ** | 0.79 ** | 0.59 ** | 0.52 ** | 0.45 ** | 0.43 ** | 0.21 ** | 0.53 ** |
| 6. Family support | | | | | - | 0.55 ** | 0.40 ** | 0.45 ** | 0.38 ** | 0.39 ** | 0.21 ** | 0.44 ** |
| 7. Other significant adults' support | | | | | | - | 0.39 ** | 0.45 ** | 0.41 ** | 0.33 ** | 0.15 * | 0.46 ** |
| 8. School support | | | | | | | - | 0.35 ** | 0.32 ** | 0.29 ** | 0.10 | 0.40 ** |
| 9. Positive identity | | | | | | | | - | 0.93 ** | 0.79 ** | 0.48 ** | 0.84 ** |
| 10. Personal power | | | | | | | | | - | 0.66 ** | 0.27 ** | 0.74 ** |
| 11. Self-esteem | | | | | | | | | | - | 0.18 * | 0.70 ** |
| 12. Sense of purpose | | | | | | | | | | | - | 0.20 ** |
| 13. Positive view of future | | | | | | | | | | | | - |

Note. * $p < 0.05$; ** $p < 0.01$.

For the correlations between the demographics and the main study variables, statistically significant but weak associations, ranging from −0.16 to 0.15 ($p < 0.05$), were observed between age and other significant adults' and school support as well as between gender and positive identity, personal power, and self-esteem, and finally, between mother's education and self-esteem (Table 2).

*3.2. Hierarchical Regression Analysis*

For the five sets of hierarchical regression analyses performed with positive identity (as a composite variable and as four dimensions) as dependent variables, the social support variable was positively and significantly associated with positive identity (β = 0.53, $p < 0.001$), personal power (β = 0.46, $p < 0.001$), self-esteem (β = 0.43, $p < 0.001$), sense of purpose (β = 0.22, $p < 0.01$), and positive view of future (β = 0.52, $p < 0.001$) (Table 3a). In Step 2, social support explained 26.8% of the variance in positive identity ($R^2$ C = 0.268) after the initial 2.9% explanation of demographic variables in Step 1 (adjusted $R^2$ = 0.029). This change in $R^2$ was significant: F (5,173) = 16.22, $p < 0.001$.

In the next five sets of hierarchical regression analyses, each of the three dimensions of social support was entered as an independent variable. Results, presented in Table 3b, show that family support, other significant adults' support, and school support entered in Step 2 explained 28.2% of the variance in positive identity ($R^2$C = 0.282). This change in $R^2$ was significant: F (7,171) = 12.18, $p < 0.001$. Moreover, adult support in all three ecological contexts (family, neighborhood, and school) was significantly and positively related to at least one dimension of positive identity (β = 0.15–0.27, $p < 0.05$). However, none of the support variables was associated with sense of purpose (Table 3b).

Finally, the five Pratt index calculations which were made to establish the relative importance of the support variables revealed that other significant adults' support emerged as the most important predictor with the highest Pratt index, accounting for 40.3% of the explained variance in positive identity. In contrast, school support explained 21.9% of the $R^2$C and was thus the least important predictor. Family support explained 38.7% of the $R^2$C. Gender was significantly associated with positive identity and its dimensions, with boys scoring higher than girls (Table 3b).

**Table 3.** (**a**) Regression analysis of positive identity, personal power, self-esteem, sense of purpose, and positive future: the role of adult social support. (**b**) Regression analysis of positive identity, personal power, self-esteem, sense of purpose, and positive future: the role of the three dimensions of adult social support.

**(a)**

| | Personal Identity | | | | Personal Power | | | | Self-Esteem | | | | Sense of Purpose | | | | Positive Future | | | |
|---|---|---|---|---|---|---|---|---|---|---|---|---|---|---|---|---|---|---|---|---|
| | *B* | *S.E.* | *β* | *p* | *B* | *S.E.* | *β* | *p* | *B* | *S.E.* | *β* | *p* | *B* | *S.E.* | *β* | *p* | *B* | *S.E.* | *β* | *p* |
| **Step 1** | | | | | | | | | | | | | | | | | | | | |
| Age | −0.10 | 0.10 | −0.08 | 0.316 | −0.14 | 0.11 | −0.09 | 0.211 | −0.07 | 0.13 | −0.04 | 0.582 | 0.10 | 0.15 | 0.05 | 0.429 | −0.21 | 0.14 | −0.12 | 0.130 |
| Gender | −0.21 | 0.10 | −0.16 | **0.029** | −0.24 | 0.11 | −0.17 | **0.022** | −0.28 | 0.13 | −0.16 | **0.033** | −0.13 | 0.14 | −0.07 | 0.376 | −0.12 | 0.14 | −0.07 | 0.364 |
| Father's education | −0.03 | 0.13 | −0.02 | 0.829 | −0.03 | 0.15 | −0.01 | 0.852 | 0.04 | 0.18 | 0.02 | 0.850 | −0.10 | 0.20 | −0.04 | 0.622 | −0.01 | 0.19 | −0.01 | 0.943 |
| Mother's education | 0.24 | 0.15 | 0.12 | 0.109 | 0.27 | 0.16 | 0.13 | 0.098 | 0.35 | 0.20 | 0.13 | 0.086 | 0.04 | 0.22 | 0.01 | 0.869 | 0.25 | 0.21 | 0.10 | 0.221 |
| **Step 2** | | | | | | | | | | | | | | | | | | | | |
| Age | −0.02 | 0.08 | −0.01 | 0.861 | −0.06 | 0.10 | −0.04 | 0.570 | 0.02 | 0.12 | 0.01 | 0.867 | 0.15 | 0.14 | 0.08 | 0.289 | −0.10 | 0.12 | −0.05 | 0.421 |
| Gender | −0.26 | 0.08 | −0.20 | **0.002** | −0.29 | 0.09 | −0.21 | **0.002** | −0.34 | 0.12 | −0.19 | **0.005** | −0.16 | 0.14 | −0.08 | 0.259 | −0.19 | 0.12 | −0.11 | 0.098 |
| Father's education | −0.01 | 0.11 | −0.01 | 0.904 | −0.01 | 0.13 | −0.01 | 0.922 | 0.05 | 0.16 | 0.02 | 0.754 | −0.09 | 0.19 | −0.04 | 0.648 | 0.01 | 0.16 | 0.00 | 0.965 |
| Mother's education | 0.16 | 0.13 | 0.08 | 0.207 | 0.20 | 0.15 | 0.09 | 0.181 | 0.26 | 0.18 | 0.10 | 0.157 | −0.01 | 0.22 | −0.01 | 0.952 | 0.15 | 0.18 | 0.06 | 0.409 |
| Support | 0.58 | 0.07 | 0.53 | **0.000** | 0.56 | 0.08 | 0.46 | **0.000** | 0.66 | 0.10 | 0.43 | **0.000** | 0.37 | 0.12 | 0.23 | **0.003** | 0.79 | 0.10 | 0.52 | **0.000** |

**(b)**

| | Personal Identity | | | | Personal Power | | | | Self-Esteem | | | | Sense of Purpose | | | | Positive Future | | | |
|---|---|---|---|---|---|---|---|---|---|---|---|---|---|---|---|---|---|---|---|---|
| | *B* | *S.E.* | *β* | *p* | *B* | *S.E.* | *β* | *p* | *B* | *S.E.* | *β* | *p* | *B* | *S.E.* | *β* | *p* | *B* | *S.E.* | *β* | *p* |
| **Step 1** | | | | | | | | | | | | | | | | | | | | |
| Age | −0.10 | 0.10 | −0.08 | 0.316 | −0.14 | 0.11 | −0.09 | 0.211 | −0.07 | 0.13 | −0.04 | 0.582 | 0.10 | 0.15 | 0.05 | 0.429 | −0.21 | 0.14 | −0.12 | 0.130 |
| Gender | −0.21 | 0.10 | −0.16 | **0.029** | −0.24 | 0.11 | −0.17 | **0.022** | −0.28 | 0.13 | −0.16 | 0.033 | −0.13 | 0.14 | −0.07 | 0.376 | −0.12 | 0.14 | −0.07 | 0.364 |
| Father's education | −0.03 | 0.13 | −0.02 | 0.829 | −0.03 | 0.15 | −0.01 | 0.852 | 0.04 | 0.18 | 0.02 | 0.850 | −0.10 | 0.20 | −0.04 | 0.622 | −0.01 | 0.19 | −0.01 | 0.943 |
| Mother's education | 0.24 | 0.15 | 0.12 | 0.109 | 0.27 | 0.16 | 0.13 | 0.098 | 0.35 | 0.20 | 0.13 | 0.086 | 0.04 | 0.22 | 0.01 | 0.869 | 0.25 | 0.21 | 0.10 | 0.221 |
| **Step 2** | | | | | | | | | | | | | | | | | | | | |
| Age | −0.02 | 0.08 | −0.01 | 0.861 | −0.03 | 0.10 | −0.02 | 0.792 | 0.03 | 0.12 | 0.02 | 0.786 | 0.15 | 0.15 | 0.08 | 0.312 | −0.05 | 0.12 | −0.03 | 0.658 |
| Gender | −0.26 | 0.08 | −0.20 | **0.002** | −0.31 | 0.09 | −0.21 | **0.001** | −0.35 | 0.12 | −0.20 | **0.004** | −0.16 | 0.14 | −0.08 | 0.269 | −0.21 | 0.12 | −0.12 | 0.066 |
| Father's education | −0.01 | 0.11 | −0.01 | 0.904 | −0.01 | 0.13 | −0.01 | 0.930 | 0.05 | 0.17 | 0.02 | 0.767 | −0.09 | 0.20 | −0.04 | 0.653 | 0.01 | 0.16 | 0.00 | 0.970 |
| Mother's education | 0.16 | 0.13 | 0.08 | 0.207 | 0.20 | 0.14 | 0.09 | 0.173 | 0.27 | 0.19 | 0.10 | 0.153 | −0.01 | 0.22 | −0.00 | 0.956 | 0.15 | 0.18 | 0.06 | 0.391 |
| Family support | 0.24 | 0.08 | 0.24 | **0.002** | 0.20 | 0.09 | 0.18 | **0.026** | 0.34 | 0.11 | 0.25 | **0.003** | 0.23 | 0.13 | 0.16 | 0.080 | 0.28 | 0.11 | 0.21 | **0.010** |
| Other significant adults' support | 0.22 | 0.07 | 0.26 | **0.001** | 0.24 | 0.08 | 0.26 | **0.002** | 0.17 | 0.10 | 0.15 | 0.078 | 0.09 | 0.11 | 0.07 | 0.433 | 0.31 | 0.09 | 0.27 | **0.001** |
| School support | 0.15 | 0.06 | 0.17 | **0.015** | 0.15 | 0.07 | 0.15 | **0.037** | 0.17 | 0.09 | 0.15 | 0.056 | 0.04 | 0.11 | 0.03 | 0.708 | 0.25 | 0.09 | 0.21 | **0.004** |

Note. *B* = Unstandardized coefficient; *S.E.* = Standard error; *β* = Standardized coefficient; *p* = Level of significance.

## 4. Discussion

The current study aimed to investigate how social support provided by the family, other significant adults, and school relate to positive identity and its dimensions in a sample of Norwegian adolescents. The findings confirmed the significant role of adult social support in boosting positive identity, adding strength to PYD literature in a new context.

In line with Hypothesis 1, the findings revealed a statistically significant association between social support and positive identity. This finding is in line with the extant PYD literature and earlier work in Italy, Romania, Norway, Slovenia, and Turkey [2,32], where external assets like support and empowerment were associated with positive identity, as an internal asset. Explaining these associations, Scales et al. [28] argued that positive developmental relationships are linked with fundamental developmental processes of a sense of identity (autonomy), a sense of agency (competence), and a commitment to community (relatedness). Accordingly, support experienced in different contexts may boost autonomy and inspire the individual to take actions and roles that will contribute to the awareness of one's motivations and behavior through developing control over one's life and future, as reflected in the assessment of positive identity.

Consistent with Hypothesis 2, the support provided by the family, other significant adults, and school was significantly related to positive identity in Norwegian adolescents. Further, the association between social support and positive adolescent identity differed across the context of support, where other significant adults' support was the strongest predictor, followed by family support and then school support. Earlier studies in Spain and Portugal [48], Hong Kong [49], and Canada [50] revealed that parental support tends to be more important to adolescents' life satisfaction than support from friends, teachers, and classmates. These previous studies partially support the current findings on Norwegian adolescents as family support was important, but not the most important, in middle and late adolescence. Nevertheless, the feeling of love, support, and safety at home and being able to seek counsel and advice from parents appeared to be important for a positive sense of identity among Norwegian adolescents, despite the individualistic culture's call for autonomy. Our findings are consistent with earlier studies that found that having responsive and loving parents who are engaged and interested in their children's life and education was a significant predictor of health, life satisfaction, and prosocial development in adolescence [34,35].

As for support experienced from non-family contexts, support from other significant adults (with the lowest mean score) was surprisingly perceived as most important, even more than family support. Support from other significant adults reflects caring neighbors and non-parental adults. Theoretically, the results can be explained by the characteristics of the adolescence period. Embedded within this period of life are many biological, social, and cognitive changes. Adolescents become more independent from their parents, thus expanding their social networks and sociocultural contexts [6]. It is possible that some students in the Norwegian context were more oriented toward relationships in the out-of-home context because this is where they spend a large amount of free time. Indeed, the emphasis on recreational, cultural, and sports practices is quite evident in youth policies in Norway [51]. Norway has a robust policy framework to provide opportunities for young people to participate in sporting, leisure, and cultural activities—and many young people do avail themselves of these opportunities.

In our study, although school support was also an important predictor of positive identity, it was the weakest of the three support dimensions. This finding may be surprising given that the school milieu is a critical developmental context in adolescence, although it could be that students do not actively seek support from teachers at school as they will naturally do with parents and other significant adults they are in contact with in their local community. Nonetheless, previous research has established the school environment as an essential source of connectedness and belonging, emerging from supportive relationships with teachers, staff members, and peers [42]. Indeed, statistics on school satisfaction show that 86% of Norwegian students in high school find their teachers caring about them. In

addition, 74% of boys and 70% of girls are satisfied with their schools [17]. When these statistics are evaluated together with the present findings, it is clear that, although school support was the weakest predictor of positive identity, students still find their schools caring and encouraging.

Additionally, the design of the questionnaire may provide further insight into the findings. School support was measured by only one item and assessed the school's caring and encouraging environment. Alternatively, other significant adults' support was measured by two items, assessing caring neighbors and support from adults other than parents. It is possible that for support from adults other than parents, participants also thought about teachers and school workers. Moreover, social support in the different ecological contexts was not significantly related to sense of purpose in the regression analysis, although two social support dimensions were weakly associated with it in correlation analysis. This may be related to methodological limitations regarding the measurement of sense of purpose (as it was measured with one item). However, it appears that to enhance this positive identity dimension, more integral social support is needed, as observed in our findings. While the findings suggest that different contexts may provide different types of support during adolescence, and adolescents may likewise desire support from different sources [52], future research may benefit from more concrete assessment of the different dimensions of social support. In addition, as the current study is based on pre-pandemic data, it will be interesting to assess how the pandemic has had an impact on youth's sources of support and positive identity.

### 4.1. Limitations and Recommendations

The present study has some limitations worth noting. First, data were collected from public high school students from two Norwegian regions, thus limiting the generalizability to the general Norwegian adolescent population. It is necessary for future research to extend data collection across larger geographic areas and to include both rural, semi-urban, and urban populations as well as students that attend private schools to secure a more representative sample. Further, due to the cross-sectional design, the models tested do not allow for any casual interpretations of the findings. For instance, it may be that adolescents who report higher positive identity are better at identifying the positive sources of support in their lives. Longitudinal studies that can investigate the directional relationship between the variables, for example, whether adult support enhances positive developmental outcomes [50] are needed in future research. Moreover, we have only studied support that adolescents can receive in-person, and not support that can be accessed through social media, which is an important youth context. In further studies, the inclusion of the support adolescents receive through this avenue may provide a more accurate perspective of the influence of social support on positive identity.

Finally, as addressed earlier, the current study focuses on one external asset category, support, and one internal asset category, positive identity. The Developmental Asset Framework consists of eight asset categories. Although each of the asset categories can be addressed independently, future research might consider examining other external assets in relation to the different dimensions of positive identity. Moreover, dimensions like school support, self-esteem, a sense of purpose, positive view of personal future and other significant adults' support that were measured with only one or two items can be further developed to improve their psychometric properties.

### 4.2. Implications for Research, Policy, and Practice

Despite the study's limitations, the findings have implications for further research, policy, and practice. The current study contributes to the PYD field by empirically confirming the role of support on different dimensions of positive identity in a sample of Norwegian adolescents. These findings highlight the need to investigate more into the context-specific sources of support that may promote positive identity in young people. For example, future research may explore a young person's access to resources, social inclusion,

leisure opportunities, and developmental programs [2]. Using qualitative methods, such as interviews and focus group discussions, adolescents' sources of support and where they encounter them can be probed into, as information on these can inform policy and practice.

As for implications for policy, because a person's sense of identity depends on interactions between self and society, identity development reflects the embodied person-in-context approach [53,54] and as such our findings provide a basis for developing policies and interventions that will offer rich and supportive social contexts to facilitate these interactions. The practical implications of our findings are that the family, school, neighborhood, and local community are important contexts to foster positive development. Thus, these contexts need to be cultivated to enhance their roles. Norway has well-established youth clubs, which are arranged outside school hours, with parents, teachers, and other responsible adults from the local community as coordinators, and usually attended by about half of the youth [55]. These club arrangements have been identified as a fundamental and effective way to cultivate a positive sense of self in adolescents [56]. However, in school settings, where adolescents spend most of the day, efforts can be made to improve the interaction between teachers and students, especially because school support was the weakest predictor of positive identity.

### 5. Conclusions

Positive identity development today, compared to previous decades, is challenging due to the variety of options available and the increased standards of the skills young people must acquire [2]. However, developing a coherent identity in adolescence is still critical for a successful transition into adulthood. The present study contributes to advancing PYD research and its implementation in the Norwegian context by confirming the relevance of adult social support for positive identity in adolescents. This study is also in line with the request for more research on positive and supportive processes instead of problem behaviors in adolescent development [1,57]. The findings support the notion that adolescent development needs to be explored and endorsed from a positive perspective that considers multiple youth ecological contexts in which they live, learn, and grow.

**Author Contributions:** Conceptualization, A.J.-T. and N.W.; methodology, A.J.-T. and N.W.; software, A.J.-T. and N.W.; validation, A.J.-T. and N.W.; formal analysis, A.J.-T. and N.W.; data curation, N.W.; writing—original draft preparation, A.J.-T.; writing—review and editing, A.J.-T. and N.W. All authors have read and agreed to the published version of the manuscript.

**Funding:** This research was supported in part by the Faculty of Psychology, University of Bergen, Norway.

**Institutional Review Board Statement:** This study was conducted in accordance with the Declaration of Helsinki and approved by the NSD–Norwegian Centre for Research Data.

**Informed Consent Statement:** Informed consent was obtained from all subjects involved in this study.

**Data Availability Statement:** The data presented in this study are available on request from the corresponding author.

**Acknowledgments:** We would like to acknowledge the Norwegian youth for their participation in this study.

**Conflicts of Interest:** The authors declare no conflict of interest.

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
