# Peer review of "Positive Youth Identity: The Role of Adult Social Support"

_2673-995X, doi:10.3390/youth3030056_

Round 1
Reviewer 1 Report
First, I enjoyed reading this manuscript as I am currently teaching a course in promoting PYD (graduate level) and have been re-reading a lot of relevant PYD literature, including publications and chapters using Search Institute's developmental assets model. When this gets published, I'm likely to assign it as reading the next summer I cover this course. I have pasted my comments and recommendations below.
Define thriving from Search Institute scholars like Benson or others in PYD practice. For example, how does positive identity as an outcome of interest (line 202, p. 4) serve as an indicator of thriving? I don’t think it’s a problem that positive identity (as an internal asset category with 4 measured asset indicators) is treated as a DV because it can be also considered an indicator of thriving and be influenced by PYD program efforts and other positive environmental influences (e.g., external asset category of support).
Line 189-90 on p. 4
Expand on Cote’s point about how cultural diversity occurs in youths’ perceptions of support. From researcher’s perspective and available literature, what might differentiate Norwegian adolescents from others (youth in US or other parts of Europe) in their perceptions of perceived support.
For Lines 217-19 on p. 5, How will understanding the differential contributions of family, other adults in their lives, and school supports to positive identity inform policy and practice?
Unique study examining how an external asset predicts an internal asset.
Results
p. 7 line 312-313 how are mean scores in social support (3.0 rounded) and positive identity (2.9) considered moderate to high when in review of literature on p. 2 (Norwegian sociocultural context section) self-esteem mentioned as low with 2.8 for females and 3.2 for males? All used same scales of 1-4.
On what is erroneously labeled as p. 2 in manuscript in results section starting with line 344, why is there no mention how sense of purpose (table 3b) as 1 of 4 indicators of positive identity was not significantly associated with any of the 3 types of social support? Nor is it mentioned in the discussion with any explanation.
I did enjoy reading the results of PRATT-index analysis (line 360—366) and how other significant adult support explained the most variance in Norwegian youths’ positive identity.
There are a couple of minor typos in the beginning of the discussion: line 369 other significant adults (or “another significant adult”), line 376 , “Scales et al.’s [28] argue”
Line 394 on “page 2” should state Norwegian adolescents not adolescence
Interesting implications described in the paragraph starting with line 398 and how adolescents are seeking out other adult support in different community contexts, including in recreational and sports contexts. I also wonder how this finding lends itself to future study examining who other significant adults are in the lives of Norwegian youths and where youth encounter them (e.g., sports contexts). I imagine a qualitative study could answer these questions. Such findings will have strong implications for policies with out of school time and youth program implementation. Lines 426-429 somewhat touch on this for future research.
Lines 437-442 are well-stated and pretty much refer to the common weakness of current PYD literature reflecting on a need for more longitudinal research. I did enjoy the point made how youth who report greater levels of positive identity may be better equipped to pinpoint sources of adult support in their lives.
Around line 465-66 when Norwegian youth clubs are mentioned (great stuff!), can more be stated about where and how they happen (sports settings? In schools outside school hours, in communities)? Are they supported by adult leadership? I think mentioning a bit more about these clubs (what they are about – how many adolescents in Norway or what proportion are involved in them) could provide insight into your findings about other adult support of positive identity.
Excellent inclusion of relevant PYD literature.
Author Response
Comments and Suggestions for Authors
Authors’ response: Thank you for your generous comments. In our revision, we have tried as much as possible to revise our manuscript in line with the different comments. Please see below for our responses.
First, I enjoyed reading this manuscript as I am currently teaching a course in promoting PYD (graduate level) and have been re-reading a lot of relevant PYD literature, including publications and chapters using Search Institute's developmental assets model. When this gets published, I'm likely to assign it as reading the next summer I cover this course. I have pasted my comments and recommendations below.
Authors’ response: Thank you very much. We are honored that our research can be used in your teaching course.
Define thriving from Search Institute scholars like Benson or others in PYD practice. For example, how does positive identity as an outcome of interest (line 202, p. 4) serve as an indicator of thriving? I don’t think it’s a problem that positive identity (as an internal asset category with 4 measured asset indicators) is treated as a DV because it can be also considered an indicator of thriving and be influenced by PYD program efforts and other positive environmental influences (e.g., external asset category of support).
Authors’ response: We have now defined thriving as suggested. Please see page 3, lines 129 – 132.
Line 189-90 on p. 4
Expand on Cote’s point about how cultural diversity occurs in youths’ perceptions of support. From researcher’s perspective and available literature, what might differentiate Norwegian adolescents from others (youth in US or other parts of Europe) in their perceptions of perceived support.
Authors’ response: Thank you for this comment. We added a few words in our revision to expand on Cote’s point about cultural diversity in youths’ perceptions of support. See page 4, lines 194 – 197.
For Lines 217-19 on p. 5, How will understanding the differential contributions of family, other adults in their lives, and school supports to positive identity inform policy and practice?
Authors’ response: Thank you. We have added some information about how understanding the differential contributions of family, school and other adults’ support can inform policy and practice. See page 5, lines 227 – 230.
Unique study examining how an external asset predicts an internal asset.
Results
p. 7 line 312-313 how are mean scores in social support (3.0 rounded) and positive identity (2.9) considered moderate to high when in review of literature on p. 2 (Norwegian sociocultural context section) self-esteem mentioned as low with 2.8 for females and 3.2 for males? All used same scales of 1-4.
Authors’ response: Thank you for pointing this out. We agree with you and have made some adjustments to the texts. Please see p. 2, lines 80 – 81 and p.7, line 323.
On what is erroneously labeled as p. 2 in manuscript in results section starting with line 344, why is there no mention how sense of purpose (table 3b) as 1 of 4 indicators of positive identity was not significantly associated with any of the 3 types of social support? Nor is it mentioned in the discussion with any explanation.
Authors’ response: Thank you for your comment. We agree with you and have added some information about the findings on sense of purpose in the results and discussion sections. Please see lines 371 – 372, and lines 443 – 449.
I did enjoy reading the results of PRATT-index analysis (line 360—366) and how other significant adult support explained the most variance in Norwegian youths’ positive identity.
There are a couple of minor typos in the beginning of the discussion: line 369 other significant adults (or “another significant adult”), line 376 , “Scales et al.’s [28] argue”
Line 394 on “page 2” should state Norwegian adolescents not adolescence
Authors’ response: Thank you for pointing these out. We have now corrected these typos as well as read through the manuscript and made other necessary corrections.
Interesting implications described in the paragraph starting with line 398 and how adolescents are seeking out other adult support in different community contexts, including in recreational and sports contexts. I also wonder how this finding lends itself to future study examining who other significant adults are in the lives of Norwegian youths and where youth encounter them (e.g., sports contexts). I imagine a qualitative study could answer these questions. Such findings will have strong implications for policies with out of school time and youth program implementation. Lines 426-429 somewhat touch on this for future research.
Authors’ response: Thank you for this suggestion. We have included some information about that in our revision. Please see lines 488 – 491.
Lines 437-442 are well-stated and pretty much refer to the common weakness of current PYD literature reflecting on a need for more longitudinal research. I did enjoy the point made how youth who report greater levels of positive identity may be better equipped to pinpoint sources of adult support in their lives.
Authors’ response: Thank you for this positive comment.
Around line 465-66 when Norwegian youth clubs are mentioned (great stuff!), can more be stated about where and how they happen (sports settings? In schools outside school hours, in communities)? Are they supported by adult leadership? I think mentioning a bit more about these clubs (what they are about – how many adolescents in Norway or what proportion are involved in them) could provide insight into your findings about other adult support of positive identity.
Authors’ response: Thank you. We agree with you and have included some information about the clubs in our revision. Please see lines 499 – 501.
Excellent inclusion of relevant PYD literature.
Authors’ response: Thank you.
Reviewer 2 Report
Thank you for this research. It shows excellent work and models the method and reporting that must be done for even more adolescent and emerging adult populations.
I have no critique of your argument or findings in that you prove what you set out to research and understand. Again, this is a masterclass in article writing!
I do wonder about a few items:
- Your concentric circles' relationships [family, school, neighbor/non-family adult] make sense, but social media or online influences are not mentioned. While, perhaps, beyond the scope of your focus, it seems impossible not to raise these influences that permeate most relationships and impact adolescent identity development. One wonders if the less favorable scores amongst females may have to do with what we see emerging in the literature about the effects of social media and females [latest APA reports, etc.].
- You note your surprise that school relationships were perceived as lower than parents or neighborhood. You admit that your survey was limited. I do wonder if lower scores may have to do with the reality that schools grade performance. Of the three, this relationship seems most conditional, where one can succeed or fail. One more expects parents to love their children unconditionally [generally], and non-family members are often chosen voluntarily [vs. School, which is required]. Perhaps more nuance could be considered.
- Data is drawn from 2019, pre-pandemic. While Scandinavian countries addressed the pandemic differently [between each other and globally], one wonders if anything has shifted regarding adolescent/em adult mental health and the support they seek, family relationships, school, neighborhood, etc.
Thank you for your work!
Author Response
Comments and Suggestions for Authors
Thank you for this research. It shows excellent work and models the method and reporting that must be done for even more adolescent and emerging adult populations.
I have no critique of your argument or findings in that you prove what you set out to research and understand. Again, this is a masterclass in article writing!
Authors’ response: Thank you for your generous comments. In our revision, we have tried as much as possible to revise our manuscript in line with the different suggestions that were made. Please see below for our responses.
I do wonder about a few items:
Your concentric circles' relationships [family, school, neighbor/non-family adult] make sense, but social media or online influences are not mentioned. While, perhaps, beyond the scope of your focus, it seems impossible not to raise these influences that permeate most relationships and impact adolescent identity development. One wonders if the less favorable scores amongst females may have to do with what we see emerging in the literature about the effects of social media and females [latest APA reports, etc.].
Authors’ response: Thank you for this comment. We agree with you and have added some information on this in our revision. Please see lines 467 – 471.
You note your surprise that school relationships were perceived as lower than parents or neighborhood. You admit that your survey was limited. I do wonder if lower scores may have to do with the reality that schools grade performance. Of the three, this relationship seems most conditional, where one can succeed or fail. One more expects parents to love their children unconditionally [generally], and non-family members are often chosen voluntarily [vs. School, which is required]. Perhaps more nuance could be considered.
Authors’ response: Thank you. We agree with your line of reasoning and have added some information about it in our revision. Please see lines 427 – 430.
Data is drawn from 2019, pre-pandemic. While Scandinavian countries addressed the pandemic differently [between each other and globally], one wonders if anything has shifted regarding adolescent/em adult mental health and the support they seek, family relationships, school, neighborhood, etc.
Authors’ response: Thank you for this comment. Your argument is plausible, and we agree with it. In our revision we have noted that it may be interesting to study how the pandemic has had an impact on youth’s sources of support and positive identity. Please see lines 452 – 454.
Thank you for your work!
Authors’ response: Thank you!